# Effectiveness and safety of low-dose versus standard-dose rivaroxaban and apixaban in patients with atrial fibrillation

Sylvie Perreault[1,2,3]*, Robert Côté[4], Alice Dragomir[1], Brian White-Guay[5], Aurélie Lenglet[6,7], Marc Dorais[8]

1 Faculty of Pharmacy, Université of Montréal, Montreal, Quebec, Canada, 2 Chaire Sanofi sur l'utilisation des médicaments, Université de Montréal, Montreal, Quebec, Canada, 3 Centre de recherche en santé publique (CReSP), partenaire CIUSSS du Centre-Sud-de-l'Île-de-Montréal et l'Université de Montréal, Montreal, Quebec, Canada, 4 Faculty of Medicine, Department of Neurology and Neurosurgery, McGill University, Montreal, Quebec, Canada, 5 Faculty of Medicine, Université of Montréal, Montreal, Quebec, Canada, 6 Faculty of Pharmacy, EA 7517, Laboratory MP3CV, Jules Verne University of Picardie, Amiens, France, 7 Department of Pharmacy, Amiens Picardie University Medical Center, Amiens, France, 8 StatSciences Inc., Notre-Dame-de-l'Île-Perrot, Quebec, Canada

* sylvie.perreault@umontreal.ca

**Data Availability Statement:** All relevant data are within the paper and its Supporting Information files.

## Abstract

### Background

Low-dose direct oral anticoagulant (DOAC) use is quite prevalent in clinical practice, but evidence of its effectiveness and safety compared with high-dose DOAC in patients with atrial fibrillation (AF) remains limited. We aimed to assess the effectiveness and safety of low-dose and high-dose DOACs in patients with AF with similar baseline characteristics.

### Methods

We used a cohort of hospitalized patients with a primary or secondary diagnosis of AF after discharge to the community, whose data were stored in the Quebec administrative databases, from 2011 to 2017. Older adults with AF newly prescribed with rivaroxaban (15 or 20 mg) or apixaban (2.5 mg or 5 mg) were classified as under treatment (UT) and intent to treat (ITT). We used an inverse probability treatment weighting study of new users of rivaroxaban and apixaban to address confounding by indication. The primary effectiveness outcome was ischemic stroke/systemic embolism (SE), while the primary safety outcome was major bleeding (MB). We used Cox proportional models to estimate the marginal hazard ratios (HRs).

### Findings

A total of 1,722 and 4,639 patients used low-dose and standard-dose rivaroxaban, respectively, while 3,833 and 6,773 patients used low-dose and standard-dose apixaban, respectively. No significant difference was observed in the incidence of comparative stroke/SE and MB between low-dose and standard-dose rivaroxaban, except for the risk of acute myocardial infarction (AMI), which was increased with the low dose in the UT analysis. For

**Funding:** This study was supported by the Heart and Stroke Foundation of Canada (G-17-0018326) and the Réseau Québécois de Recherche sur les Médicaments (RQRM). Please refer to the following URL: https://www.heartandstroke.ca/ and https://www.rqrm.ca/. The information contained in this manuscript has not been published previously and is not under review for publication elsewhere. The study findings were presented with posters and abstracts at the International Society for Pharmacoeconomics and Outcomes Research (ISPOR) 24th Annual International Meeting (December 1–3, 2021) and 37th International Conference on Pharmacoepidemiology and Therapeutic Risk Management (ISPE) (August 21–25, 2021), both of which were funded by Heart and Stroke Foundation and RQRM. MD, the sole founder and representative of StatSciences Inc., provided us with independent biostatistics consultancy. These funders provided support in the form of salaries for author MD, " and the remaining authors declare that the research was conducted in the absence of any commercial or financial relationships that could be construed as a potential conflict of interest ". The specific roles of these authors are articulated in the 'author contributions' section. All funders had no role in study design, data collection and analysis, decision to publish, or preparation of the manuscript.

**Competing interests:** The authors have declared that no competing interests exist.

apixaban, no difference was found in the bleeding rates, but the risk of stroke/SE (HR: 1.95; 95% confidence interval (CI): 1.38–2.76) and death (HR: 1.99; 95% CI: 1.46–2.70) were greater in the low-dose group than in the standard-dose group in the UT analysis. Similar results were observed for the ITT analysis.

## Conclusion

No significant differences were observed in the effectiveness or safety outcome between low-dose and standard-dose rivaroxaban, except for AMI. However, low-dose apixaban was associated with a greater risk of stroke/SE and death without a reduction in the bleeding rates.

## Introduction

Atrial fibrillation (AF) is a common cause of embolic stroke, especially in the older population. In the coming years, its prevalence will likely increase [1]. Ischemic stroke associated with AF is more severe and has a higher mortality rate [2]. Oral anticoagulant (OAC) therapy is effective in preventing ischemic events including strokes in patients with non-valvular AF [3–6]. The use of warfarin therapy increases in older adults, given that age is an independent risk factor for both thromboembolic and bleeding complications [6–9]. The limitations of warfarin have led to the widespread use of direct OACs (DOACs), which has a lower risk of drug and dietary interactions. DOACs are proven alternative therapies to warfarin for the prevention of stroke and systemic embolism (SE) in patients with non-valvular AF [3, 5, 6].

DOACs as treatment for non-valvular AF have been evaluated in large randomized clinical trials (RCTs). These studies have shown DOACs to be comparable to warfarin in terms of efficacy, with similar or reduced rates of major bleeding, especially intracranial hemorrhage [10–13]. Recent population-based studies have examined the dosage of DOACs administered for AF [14, 15]. These studies showed that a greater proportion of patients received a lower dose of DOACs, which is in contrast to the report of previous RCTs [14, 15]. The greater use of low-dose DOACs may be associated with higher risk of myocardial infarction, other ischemic events and death, whereas use of standard doses may be associated with a higher risk of gastro-intestinal bleeding [16, 17]. Based on real-world studies, standard-dose apixaban presents a better benefit-risk profile than rivaroxaban [18]. However, studies reporting on the safety and effectiveness of reduced doses of DOAC compared with standard doses are limited [17, 19]. A recent patient-level network meta-analyses of RCTs of DOACs versus warfarin in patients with AF reported results that compared to lower-dose DOAC, patients randomized to standard-dose DOAC presented a lower risk of stroke/SE (hazard ratio (HR): 0.76; 95% confidence interval (CI): 0.68–0.86) and of composite efficacy outcome (HR: 0.89; 95% CI: 0.82–0.96). However, this study presents limitations such as, drug adherence was not taken into consideration and combining RCTs with different study drugs and doses for meta-analyses may not reveal the difference between outcomes that are specific each DOAC [20].

Given that the use of low-dose DOAC is more prevalent in clinical practice, and that close to 50% of patients who receive the low-dose apixaban do not meet the least two of three clinical characteristics [21], it is important to assess the effectiveness and safety of low-dose vs standard-dose DOACs used in AF patients with similar baseline characteristics [14, 15]. Due to the lack of randomization in treatment assignment, we used an inverse probability treatment weighting (IPTW) approach to minimize the impact of confounding by indication in

representative cohorts of AF patients to assess the effectiveness and safety of low-dose vs standard-dose rivaroxaban and apixaban [22].

## Methods

### Database

We included a cohort from the Med-Echo administrative databases, which store data on hospital discharges, medical services, and public drug plans, managed by the Régie de l'Assurance Maladie du Québec (RAMQ) (S1 Table) [23–26]. The databases were linked using encrypted health insurance numbers. The information from these databases provide a comprehensive picture of the status of hospital admissions. The protocol was approved by the University of Montreal Ethics Committee.

### Study population

We conducted the analysis using administrative data from 16,967 new users of rivaroxaban and apixaban. We identified adult patients (aged 18 years and older) who were hospitalized for all causes from January 1, 2011 to December 31, 2017, and were discharged alive to the community with inpatient coding for AF as a primary or secondary diagnosis using International Classification of Diseases (ICD)-9 (427.3, 427.31, or 427.32) or ICD-10 (I48) codes [27, 28]. For patients with more than one eligible hospital admission with a diagnosis of AF, we used the date of the first admission as the eligible date. ICD-9 coding to identify AF performed relatively well in previous validation studies, with a median positive predictive value of over 80% [29].

We identified patients who filled a new prescription for rivaroxaban (low-dose 15 or standard-dose 20 mg once daily) or apixaban (low-dose 2.5 or standard-dose 5.0 mg twice daily) within a year after hospital discharge. The patients were new users, defined as no exposure to any OAC one year prior to the claim index date. Eligible patients were required to have pharmacy coverage for at least 12 months and had been continually enrolled in an insurance drug plan for at least one year before the claim index date which defined the cohort entry. Patients prescribed with either dabigatran or edoxaban were not included because of their relatively small number.

We excluded patients who had undergone cardiac valvular replacement or valvular procedures within five years of cohort entry. We excluded patients who were diagnosed with end-stage chronic kidney disease, those who had undergone kidney transplantation, those who were on dialysis in a 3-year period prior to cohort entry, and those who were diagnosed with deep vein thrombosis or who had undergone orthopedic surgery within three months prior to cohort entry. Finally, we excluded patients with a coagulation deficiency or those who had undergone certain medical procedures, including cardiac catheterization, stent, coronary artery bypass grafting, cerebrovascular interventions, or cardiac device implantation within three months of cohort entry.

### Exposure ascertainment

We established rivaroxaban and apixaban treatment periods using the fill dates and days that the medications were supplied per prescription. Patients were categorized as under treatment (UT) if they filled prescriptions within 30 days after the end of the last treatment period. We also performed an intent-to-treat (ITT) analysis in which the censoring criteria for drug discontinuation or switching were not applied.

## Outcome ascertainment

The primary effectiveness outcome was ischemic stroke/SE. We also assessed the rate of other specific outcomes, such as acute myocardial infarction (AMI) and all-cause mortality. The primary safety outcome was major bleeding, including intracranial hemorrhage (ICH), gastrointestinal bleeding, and major bleeding, from other sites. We also assessed the rates of each category of major bleeding. Major bleeding was defined as previously published [28]. We identified the outcomes using ICD-9 or ICD-10 codes to identify the primary diagnosis of inpatient claims (S2 Table). The positive predictive value was >80% [30, 31]. These codes performed relatively well in previous validation studies [32, 33].

## Patient demographics and clinical characteristics

We documented the patients' demographics at cohort entry. We identified the associated comorbidities during hospitalization as well as those that occurred within the three years prior to the cohort entry [28, 30, 34]. The $CHA_2DS_2$-VASc score was determined based on the patients' characteristics and associated comorbidities (S2 and S3 Tables). We determined the modified HAS-BLED score for each patient (S4 and S5 Tables) [35–37]. The Charlson-Deyo Comorbidity Index was used to assess the level of comorbidity [38, 39]. The frailty score within two years prior to the cohort entry was also evaluated using the adapted elders risk assessment (ERA) index [40, 41]. The category of chronic kidney disease was assessed using the estimated glomerular filtration rate using an algorithm based on diagnosis code, drug use, and nephrologist visits from administrative databases that shown to be valid when compared with medical chart reviews in older adults. The algorithm used for eGFR definition had a positive predictive value ranging from 94.5% to 97.7% [34]. Finally, we assessed all prescriptions filled for different medications within two weeks prior to cohort entry and the significant drug interactions (S6 Table).

## Statistical analyses

Descriptive statistics were used to summarize the demographic and clinical characteristics of the patients according to the type of DOAC initially used after hospital discharge.

To balance the distribution of patient characteristics between groups, an IPTW method was employed [22, 42]. We created two IPTW populations: 1) low- and standard-dose rivaroxaban IPTW population and 2) low- and standard-dose apixaban IPTW population. We used a multivariable logistic regression model to estimate the probabilities (propensity score (PS)) of being in the treatment group (low dose) actually observed, based on all baseline covariates. IPTW was calculated using PS. Weighing patients based on the inverse of this conditional probability allows the establishment of a pseudo-population and creates a balance across treatment groups in terms of the covariates included in the model (S7 and S8 Tables). The IPTW approach attempts to minimize the impact of confounding bias in observational studies by approximating a randomization process used in RCTs. All weights were stabilized by multiplying the IPTW by the marginal probability of being in the treatment group.

Descriptive statistics of each IPTW population were used to characterize the patients. We estimated the standardized differences in baseline characteristics between the treatment groups; a difference of >10% may suggest a meaningful imbalance [42]. For descriptive analyses, we presented the pre- and post-match between-group comparisons. We reported the outcomes per 100 person-years for each treatment in each IPTW population.

Patients were followed from the index date until the earliest date of occurrence of the following events: outcome, being institutionalized or hospitalized for greater than 15 days, end of enrollment in drug plans, or death, whichever came first. We compared the two IPTW

populations based on the treatment provided (standard-dose rivaroxaban or apixaban as reference) to estimate the marginal Cox hazard ratios (HRs) for outcomes, under respectively UT and ITT analysis.

### Sensitivity analyses

First, the sensitivity analyses of the negative control are presented in S11 Table. Second, we calculated the expect value (E-value) to assess the impact of unmeasured confounders [43] (S12 Table). The E-value indicates the strength of the association between an unmeasured confounder and the use of both low- and standard-dose DOACs, and the outcomes to reduce the observed effect to the null, based on the measured covariates. The distribution of follow-up, level of adherence and sample size estimation are presented at S13 and S14 Tables.

Third, we used 1:1 PS matching without replacement and match on the logit of the PS using calipers of width equal to 0.2 of the standard deviation of the logit of the PS [44]. We used a multivariable logistic regression model to estimate the PS of the patients receiving low vs standard dose DOACs, based on baseline characteristics (S15 Table). All analyses were performed using SAS statistical software (version 9.4; SAS Institute Inc., Cary, NC, USA).

## Results

### Demographic and clinical characteristics

Fig 1 shows the cohort selection process. A total of 16,967 qualifying AF patients received a claim for rivaroxaban or apixaban: 1,722 for low-dose rivaroxaban, 4,639 for standard-dose rivaroxaban, 3,833 for low-dose apixaban, and 6,773 for standard-dose apixaban.

The characteristics of the study population are shown in Table 1 based on the index drug used after incorporating IPTW. In the rivaroxaban groups, the patients' mean age was 75.7–75.9 years, and 49.5–49.8% were women; in the apixaban groups, the patients' mean age was 79.8–80.2 years, and 55.8–56.1% were women. As shown in S7 and S8 Tables, the absolute standardized baseline differences in the initial cohort were well balanced after IPTW population. The Kaplan-Meier curves of the clinical outcomes of low- and standard-dose rivaroxaban and apixaban as UT during follow-up are shown in S1 and S2 Figs.

### Effectiveness and safety outcomes under UT and ITT

Pairwise comparisons of low-dose and standard-dose rivaroxaban adjusted using IPTW for sociodemographic, comorbidities, concomitant drugs, and healthcare use are shown in Table 2 (S9 and S10 Tables). No significant difference was observed in the HRs for the primary effectiveness outcome (ischemic stroke/SE) (HR: 1.16; 95% CI: 0.70–1.93 and HR: 1.28; 95% CI: 0.83–1.99) and safety outcome for UT and ITT (HR: 0.98; 95% CI: 0.69–1.41; HR: 1.01; 95% CI: 0.72–1.41) between low-dose rivaroxaban and standard-dose rivaroxaban. No significant difference was observed in all other outcomes, except for an increased risk of AMI associated with the lower dose in the UT analysis (HR: 2.07; 95% CI: 1.21–3.52).

Results of the pairwise comparisons of low-dose and standard-dose apixaban are shown in Table 3 (S9 and S10 Tables). Low-dose apixaban was associated with a greater risk for ischemic stroke/SE, all-cause death, and effectiveness composite events than standard-dose apixaban (HR: 1.95; 95% CI: 1.38–2.76; HR: 1.99; 95% CI: 1.46–2.70; HR: 1.74; 95% CI: 1.41–2.13, respectively), without any significant decrease in the incidence of safety composite events in UT analysis (HR: 0.76; 95% CI: 0.56–1.02). Similar results were observed for the ITT analysis.

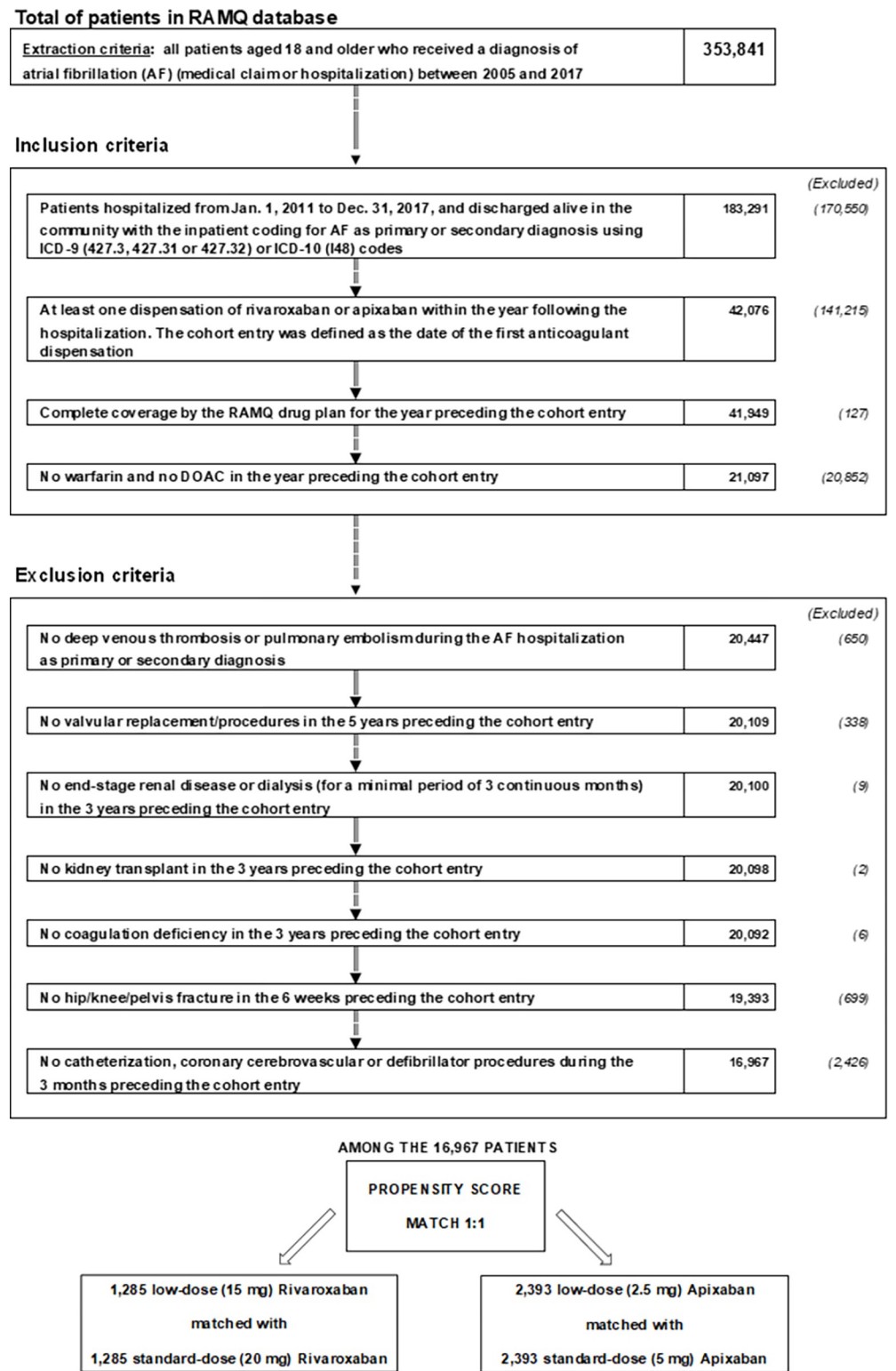

**Fig 1. Flow chart of the study design and patient selection process.** AF: atrial fibrillation; OAC: oral anticoagulant; RAMQ: Régie d'Assurance Maladie du Québec (Quebec administrative databases).

**Table 1. Baseline characteristics of the atrial fibrillation populations after inverse probability of treatment weighting.**

| | IPTW rivaroxaban population | | IPTW apixaban population | |
|---|---|---|---|---|
| | **Low dose Rivaroxaban (n = 1,722)** | **Standard dose Rivaroxaban (n = 4,639)** | **Low dose Apixaban (n = 3,833)** | **Standard dose Apixaban (n = 6,773)** |
| **Age**, mean (SD) | 75.7 (9.7) | 75.9 (9.8) | 79.8 (9.3) | 80.2 (9.4) |
| **Female sex (%)** | 49.5% | 49.8% | 55.8% | 56.1% |
| **CHA$_2$DS$_2$-VASc score**, mean (SD) | 3.2 (1.5) | 3.3 (1.5) | 3.8 (1.3) | 3.8 (1.4) |
| **HAS-BLED score**, mean (SD) | 2.7 (1.3) | 2.7 (1.3) | 3.2 (1.3) | 3.1 (1.3) |
| **Charlson Comorbidity Index**, mean (SD) | 4.2 (3.4) | 4.0 (3.4) | 4.7 (3.4) | 4.6 (3.5) |
| **Frailty index**, mean (SD) | 10.4 (6.7) | 10.0 (6.6) | 11.8 (7.0) | 11.8 (7.3) |
| **Comorbidities ¥ (%)** | | | | |
| Hypertension | 75.8% | 77.3% | 82.2% | 80.7% |
| Dyslipidemia | 46.4% | 49.5% | 54.6% | 52.6% |
| Diabetes | 33.5% | 31.7% | 35.9% | 33.8% |
| Coronary artery disease | 47.2% | 46.4% | 49.9% | 49.2% |
| Acute myocardial infarction | 11.1% | 10.9% | 13.7% | 13.9% |
| Chronic heart failure | 30.0% | 30.3% | 38.0% | 36.8% |
| Cardiomyopathy | 5.3% | 6.1% | 6.1% | 5.6% |
| Other dysrhythmias | 18.6% | 18.0% | 18.3% | 18.7% |
| Valvular disease | 14.0% | 14.9% | 18.0% | 18.0% |
| Prior cerebrovascular disease including TIA | 13.1% | 15.5% | 20.3% | 19.3% |
| Prior ischemic stroke | 12.9% | 15.1% | 19.8% | 18.7% |
| Peripheral artery disease | 18.9% | 17.8% | 20.1% | 19.5% |
| Chronic renal failure | 27.5% | 24.5% | 37.2% | 35.9% |
| Chronic renal failure <30 mL/min | 1.4% | 2.0% | 3.1% | 2.8% |
| Acute renal failure | 18.1% | 14.8% | 25.0% | 23.1% |
| Chronic obstructive pulmonary disease/asthma | 37.5% | 36.5% | 34.2% | 36.0% |
| Liver disease | 3.1% | 2.1% | 2.3% | 2.0% |
| Systemic embolism | 2.4% | 1.9% | 1.9% | 2.0% |
| Depression | 10.5% | 10.9% | 12.0% | 11.5% |
| Hypothyroidism | 23.3% | 21.5% | 23.4% | 23.6% |
| Neurologic disorder | 22.8% | 22.8% | 26.4% | 26.7% |
| Prior major bleeding | 25.4% | 23.2% | 33.0% | 30.1% |
| Malignant cancer | 27.6% | 25.6% | 26.8% | 26.3% |
| **Medical procedures (three years prior to the claim index date) (%)** | | | | |
| Cardiac catheterization | 3.6% | 3.2% | 3.2% | 3.6% |
| Percutaneous coronary intervention–Stent | 2.2% | 2.3% | 2.5% | 2.3% |
| Coronary artery bypass grafting | 0.5% | 0.6% | 0.2% | 0.6% |
| Implantable cardiac device | <0.1% | <0.1% | 0.0% | 0.0% |
| **Medications used within two weeks prior the claim index date (%)** | | | | |
| Diuretics | 34.3% | 32.8% | 38.5% | 37.5% |
| Loop diuretics | 26.1% | 25.1% | 31.9% | 30.9% |
| B-Blockers | 62.9% | 63.1% | 61.5% | 64.3% |
| Inhibitors of renin-angiotensin system | 33.5% | 35.0% | 36.3% | 35.8% |
| Calcium channel blockers | 33.7% | 35.2% | 36.9% | 36.8% |
| Statin | 40.8% | 41.2% | 42.2% | 43.4% |
| Antidiabetics | 18.5% | 17.7% | 20.1% | 20.1% |
| Antiplatelet excluding low dose ASA | 3.2% | 3.7% | 4.4% | 4.4% |
| Low dose ASA | 24.0% | 23.8% | 24.4% | 24.2% |

*(Continued)*

**Table 1.** (Continued)

| | IPTW rivaroxaban population | | IPTW apixaban population | |
|---|---|---|---|---|
| | **Low dose Rivaroxaban (n = 1,722)** | **Standard dose Rivaroxaban (n = 4,639)** | **Low dose Apixaban (n = 3,833)** | **Standard dose Apixaban (n = 6,773)** |
| Proton pump inhibitors | 36.8% | 35.0% | 40.8% | 39.4% |
| NSAIDs | 1.0% | 1.5% | 1.3% | 1.2% |
| Amiodarone or propafenone | 8.7% | 9.0% | 9.2% | 9.8% |
| Digoxin | 8.7% | 9.8% | 9.0% | 8.9% |
| Antidepressant, SSRIs | 8.4% | 7.2% | 9.7% | 8.9% |
| PGP inhibitor use | 53.3% | 54.1% | 57.3% | 57.1% |
| Strong dual inhibitors of CYP3A and PGP for rivaroxaban[‡] | 0.8% | 0.9% | - | - |
| Strong dual inducers of CYP3A and PGP for rivaroxaban[¥] | 1.0% | 0.7% | - | - |
| Strong dual inhibitors of CYP3A4 and PGP for apixaban[*] | - | - | 0.4% | 0.5% |
| Strong dual inducers of CYP3A4 and PGP for apixaban[†] | - | - | 0.6% | 0.5% |
| Number of distinct AHFS classes, mean (SD) | 8.2 (4.8) | 8.0 (4.4) | 8.7 (4.3) | 8.7 (4.3) |
| **Health medical service within 1 year prior to the claim index date** | | | | |
| Number of specialty visits, mean (SD) | 1.2 (2.1) | 1.2 (2.3) | 1.3 (2.6) | 1.3 (2.6) |
| Number of family physician visits, mean (SD) | 1.1 (2.4) | 1.0 (2.5) | 1.1 (2.6) | 1.2 (3.6) |
| Number of emergency visits, mean (SD) | 2.9 (2.2) | 2.9 (2.6) | 3.1 (2.4) | 3.1 (2.6) |
| **Health hospital service in 3-year prior the index claim** | | | | |
| Number of all-cause hospital admission, mean (SD) | 2.2 (1.8) | 2.1 (1.7) | 2.3 (1.7) | 2.2 (2.0) |
| Length of stay, mean (SD) | 8.1 (9.3) | 8.0 (35.0) | 9.1 (10.1) | 9.3 (11.3) |

[*] Strong dual inhibitors of CYP3A4 and PGP for apixaban: ketoconazole, itraconazole, ritonavir, and clarithromycin

[†]strong dual inducers of CYP3A4 and PGP for apixaban: rifampin, carbamazepine, and phenytoin

[‡]strong dual inhibitors of CYP3A and PGP for rivaroxaban: ketoconazole and ritonavir

[¥]strong dual inducers of CYP3A and PGP for rivaroxaban: rifampin, carbamazepine, and phenytoin

IPTW: inverse probability of treatment weighting, ASA: acetyl salicylic acid, NSAIDs: nonsteroidal anti-inflammatory drugs, SD: standard deviation, TIA: transient ischemic stroke, PGP: P-glycoprotein, SSRIs: selective serotonin reuptake inhibitors

## Sensitivity analyses

As shown in S11 Table, the analyses of negative controls are presented. As shown in S12 Table, the E-value closest to bound 1 for the incidence of ischemic stroke/SE events between low-dose and standard-dose apixaban was 2.43. This finding indicates that the HR of this primary effectiveness outcome could be explained by the unmeasured confounder that occurred 2.4 times more common in patients receiving low-dose apixaban than in those receiving standard-dose apixaban, and increases the rate of primary effectiveness events by 2.43 folds. Thus, a high E-value indicates that these significant results are robust to unmeasured confounding factors.

We also assess a 1:1 PS matching without replacement on the outcomes for UT and ITT analyses. Pairwise comparisons of low-dose and standard-dose rivaroxaban or low-dose and standard-dose apixaban matched for sociodemographic and clinical characteristics are shown in S15 Table. Similar results were observed as those observed using IPTW method (S16 Table). Moreover, the results were also similar to those using adjusted raw data for UT analyses (S17 Table).

## Discussion

To our knowledge, this is the first observational study to examine the comparative effectiveness and safety of low-dose vs standard-dose rivaroxaban and apixaban with the probability of

**Table 2. Comparative effectiveness and safety of low-dose and standard-dose rivaroxaban after inverse probability of treatment weighting.**

|  | Analysis | Rate per 100 PY* (95% CI) | Rate per 100 PY* (95% CI) | HR (95% CI) | P value |
|---|---|---|---|---|---|
| **Rivaroxaban** |  | **15 mg once daily (N = 1,722)** | **20 mg once daily (N = 4,639)** |  |  |
| **Effectiveness** |  |  |  |  |  |
| Stroke (ischemic only)/SE | UT | 1.8 (1.0–2.5) | 1.5 (1.1–1.9) | 1.16 (0.70–1.93) | 0.5604 |
|  | ITT | 2.0 (1.3–2.7) | 1.5 (1.2–1.9) | 1.28 (0.83–1.99) | 0.2660 |
| All-cause mortality | UT | 1.7 (1.0–2.4) | 2.4 (1.9–2.9) | 0.68 (0.42–1.11) | 0.1246 |
|  | ITT | 6.6 (5.2–7.9) | 6.7 (5.9–7.5) | 0.98 (0.78–1.24) | 0.8517 |
| Acute myocardial infarction | UT | 1.9 (1.2–2.7) | 0.9 (0.6–1.2) | 2.07 (1.21–3.52) | 0.0077 |
|  | ITT | 1.8 (1.1–2.4) | 1.5 (1.1–1.8) | 1.19 (0.75–1.89) | 0.4627 |
| Effectiveness composite | UT | 5.4 (4.0–6.7) | 4.8 (4.0–5.5) | 1.11 (0.83–1.48) | 0.4864 |
|  | ITT | 10.2 (8.5–11.8) | 9.1 (8.2–10.0) | 1.11 (0.92–1.34) | 0.2859 |
| **Safety** |  |  |  |  |  |
| Intracranial bleeding | UT | 0.4 (0.0–0.7) | 0.6 (0.3–0.8) | 0.65 (0.23–1.81) | 0.4066 |
|  | ITT | 0.4 (0.1–0.8) | 0.5 (0.3–0.7) | 0.88 (0.36–2.13) | 0.7711 |
| GI bleeding | UT | 1.4 (0.8–2.1) | 1.7 (1.3–2.2) | 0.83 (0.48–1.42) | 0.4928 |
|  | ITT | 1.3 (0.7–1.9) | 1.7 (1.3–2.1) | 0.80 (0.48–1.33) | 0.3860 |
| Extracranial bleeding | UT | 3.1 (2.1–4.1) | 2.9 (2.3–3.4) | 1.05 (0.72–1.54) | 0.7955 |
|  | ITT | 2.8 (1.9–3.7) | 2.7 (2.2–3.2) | 1.04 (0.72–1.49) | 0.8477 |
| Safety composite | UT | 3.4 (2.4–4.5) | 3.5 (2.8–4.1) | 0.98 (0.69–1.41) | 0.9289 |
|  | ITT | 3.3 (2.3–4.2) | 3.2 (2.7–3.8) | 1.01 (0.72–1.41) | 0.9481 |

*PY: person-years

†composite of benefit/risk

‡irreversible events: defined as a composite of ischemic stroke, hemorrhagic stroke, intracranial hemorrhage, acute myocardial infarction, and all-cause mortality

HR: hazard ratio, CI: confidence interval, UT: under treatment, ITT: intent to treat, SE: systemic embolism, GI: gastrointestinal

approximating a randomization process similar to RCT. In the population studied, we found that low-dose apixaban, compared with the standard-dose is associated with a greater risk for ischemic stroke/SE, all-cause death, and effectiveness composite events without any significant decrease in the safety composite events as shown in both the UT and ITT analyses. Conversely, no significant difference was observed in the effectiveness and safety outcomes based on both the UT and ITT analyses for low-dose rivaroxaban compared with the standard-dose, except for the risk of AMI, which was increased for the lower dose in the UT analysis.

Several reasons may explain these results. First, the impact of pharmacokinetic parameters should be considered. For instance, the drug plasma concentrations of DOACs in the real-world setting present a higher individual variability than those in RCTs [45]. The results of phase III trial post-hoc analyses revealed a relationship between DOAC plasma levels and thrombotic and bleeding complications in AF patients.[12, 46] But, in the Testa study, thrombotic complications only occurred in AF patients primarily treated with low-dose DOAC and those with a high $CHA_2DS_2$-VASc score with a very low C-trough level [47]. Second, the real-world use of DOACs may explain the differences observed, indicating the possibility of suboptimal use. A recent study conducted in patients during time of no-use of any OAC compared to time on warfarin reported a two- to three-fold higher risk of stroke while the bleeding risk were similar to 44% lower, and the risk of death was similar or was 44% or higher [48]. Similar risk was also reported in the Rocket study after discontinuing treatment [11]. Third, the off-label use can also be a contributing factor. In an Israeli cohort study of AF patients, 39% of those receiving a reduced off-label dose of anticoagulant agents reported a reduction in their effectiveness (composite outcome of all-cause mortality, stroke, and AMI) without a decrease

**Table 3. Comparative effectiveness and safety of low-dose and standard-dose apixaban after inverse probability of treatment weighting.**

| | Analysis | Rate per 100 PY* (95% CI) | Rate per 100 PY* (95% CI) | HR (95% CI) | P value |
|---|---|---|---|---|---|
| **Apixaban** | | **2.5 mg twice daily N = 3,833** | **5.0 mg twice daily N = 6,773** | | |
| **Effectiveness** | | | | | |
| Stroke (ischemic only)/SE | UT | 2.4 (1.8–2.9) | 1.2 (0.9–1.5) | 1.95 (1.38–2.76) | 0.0002 |
| | ITT | 2.3 (1.7–2.8) | 1.3 (1.0–1.6) | 1.68 (1.22–2.32) | 0.0016 |
| All-cause mortality | UT | 3.0 (2.4–3.7) | 1.5 (1.2–1.9) | 1.99 (1.46–2.70) | <0.0001 |
| | ITT | 11.3 (10.2–12.5) | 7.0 (6.4–7.7) | 1.61 (1.40–1.85) | <0.0001 |
| Acute myocardial infarction | UT | 1.2 (0.8–1.7) | 1.0 (0.7–1.3) | 1.21 (0.79–1.86) | 0.3845 |
| | ITT | 1.2 (0.8–1.6) | 1.3 (1.0–1.5) | 0.94 (0.64–1.40) | 0.7691 |
| Effectiveness composite | UT | 6.5 (5.5–7.4) | 3.7 (3.2–4.2) | 1.74 (1.41–2.13) | <0.0001 |
| | ITT | 14.4 (13.1–15.7) | 9.3 (8.6–10.1) | 1.53 (1.35–1.74) | <0.0001 |
| **Safety** | | | | | |
| Intracranial bleeding | UT | 0.5 (0.2–0.7) | 0.7 (0.5–0.9) | 0.69 (0.37–1.28) | 0.2354 |
| | ITT | 0.5 (0.3–0.8) | 0.7 (0.5–1.0) | 0.73 (0.42–1.28) | 0.2703 |
| GI bleeding | UT | 1.1 (0.7–1.5) | 1.0 (0.7–1.3) | 1.10 (0.70–1.72) | 0.6735 |
| | ITT | 1.1 (0.7–1.5) | 1.0 (0.7–1.2) | 1.08 (0.71–1.65) | 0.7074 |
| Extracranial bleeding | UT | 1.7 (1.2–2.2) | 2.2 (1.8–2.6) | 0.79 (0.56–1.10) | 0.1614 |
| | ITT | 1.9 (1.4–2.3) | 2.2 (1.8–2.5) | 0.86 (0.63–1.18) | 0.3490 |
| Safety composite | UT | 2.2 (1.7–2.8) | 2.9 (2.4–3.3) | 0.76 (0.56–1.02) | 0.0709 |
| | ITT | 2.4 (1.8–2.9) | 2.9 (2.5–3.3) | 0.82 (0.62–108) | 0.1504 |

*PY: person-years, HR: hazard ratio, CI: confidence interval, UT: under treatment, ITT: intent to treat, SE: systemic embolism, GI: gastro-intestinal

in bleeding rates [17]. In the ORBIT-AF II registry, patients receiving inappropriate reduced doses of DOACs had higher unadjusted incidence of stroke/SE (HR: 1.56; 95% CI: 0.92–2.67) and death (HR: 2.61; 95% CI: 1.86–3.67) [49], however, after adjustment, the outcomes were not significant but still tended towards an increased risk with the lower inappropriate doses, particularly for death (HR: 1.40; 95% CI: 0.97–2.00) [49].

Again, close to 50% of patients who receive the low-dose apixaban do not meet the least two of three clinical characteristics [21]. The higher risk profile of AF patients in real-life studies than those seen in RCTs, and the higher rates of major bleeding and mortality not attributable to thromboembolism in AF patients treated with apixaban 2.5 compared to 5 mg twice daily seem to be important causes to the higher than expected thromboembolic event rates in clinical [21]. Conversely, we could not further discuss the off-label use of these agents due to the lack of clinical data. In addition, the increased risk of AMI in patients treated with low-dose rivaroxaban could be related to the lack of antiplatelet therapy in eligible patients or to the difference in the amount of drug absorbed between the 15 or 20 mg tablets of rivaroxaban; for example, when rivaroxaban is taken on an empty stomach, the absorption of this drug is decreased, thus reducing its effectiveness.

The predictable effects of DOACs can be determined based on the anticoagulant activity or drug concentration in selected populations with different baseline risks for thrombosis, concomitant use of antiplatelet agents, or drug interactions in RCTs [45]. There is also a need to assess the plasma levels and factor Xa inhibition related to the dosage and outcomes, since the net benefit seems to vary across DOACs [47, 50]. Moreover, there are important knowledge gaps regarding DOAC dosing for clinicians treating AF, and also significant difference in dose preferences between clinicians and patients [51].

The strengths of our study include the large sample size and the comparative effectiveness and safety profile of low-dose and standard-dose DOACs; moreover, our study was performed

in a well-characterized Canadian population-based cohort. The IPTW method was applied considering several variables. Sensitivity analyses were performed, to assess the impact using 1:1 PS-Match study design, to determine the negative control and the E-value corresponding to the confidence bound closest to 1; the corresponding HR point estimate indicated that the results are robust to unmeasured confounders.

However, our study has several limitations. First, this is an observational study using administrative data and may be subject to confounding bias by unadjusted factors (e.g., AF severity, blood pressure control, exact estimated glomerular filtration rate, international normalized ratio, body weight, and over-the-counter prescriptions) or by a residual channeling bias. Second, we are not able to provide the appropriateness of the prescription without having the exact weight and the exact estimated glomerular filtration rate but in order to assess the robustness of the results, we ran several sensitivity analyses. Third, PS-Match and IPTW method have different interpretations. In the case where there are an adequate sample size of comparators for matching to treated patients, PS-Match offers a more clear method. PS-Match may also be more robust to misspecification of the propensity score than the IPTW method, where extreme weights can bias the estimation of the treatment effect. Although, IPTW has its advantages in retaining all eligible patients, which may be preferred if there are limitations in terms of sample size where PS-Match may limit the representativeness of the study population and generalizability of the overall study findings [22]. Four, our results may not be generalizable to other groups such as non-hospitalized individuals with AF and other ethnic groups, as our population was mostly white [52]. Five, our study is mainly representative of older adults; the results therefore may not be extrapolated to younger patients, although the prevalence of AF in younger patients is less common. Sixfth, the increased stroke/SE and mortality observed in the low-dose apixaban group, which was not observed in the low-dose rivaroxaban group, warrant further investigation as to whether these results reflect a true association, biases from selective prescribing, or the impact of pharmacokinetic parameters. Furthermore, we were unable to assess the causes of mortality in more detail. Finally, residual bias is still possible, especially with regard to the unmeasured variables related to the severity of disease and their effect in the healthy population.

## Conclusion

In this population of new users of DOACs as treatment for AF, low-dose apixaban presented a greater risk of stroke/SE and mortality without any positive impact on the safety profile compared with the standard-dose of apixaban. Moreover, no significant differences were found in the effectiveness or safety between low-dose and standard-dose rivaroxaban, except for the risk of AMI, which was increased with the lower dose in the UT analysis. Further research is needed to better understand the factors related to the balance between benefit and risk related to varying dosages of DOACs.

## Supporting information

**S1 Fig. Rivaroxaban 15 mg vs rivaroxaban 20 mg; analysis of the effectiveness outcomes in the under treatment cohort after inverse probability of treatment weighting.**
(TIF)

**S2 Fig. Rivaroxaban 15 mg vs rivaroxaban 20 mg; analysis of the safety outcomes of the under treatment cohort after inverse probability of treatment weighting.**
(TIF)

**S3 Fig. Apixaban 2.5 mg vs apixaban 5 mg; analysis of the effectiveness outcomes of the under treatment cohort after inverse probability of treatment weighting.**
(TIF)

**S4 Fig. Apixaban 2.5 mg vs apixaban 5 mg; analysis of the safety outcomes in the under treatment cohort after inverse probability of treatment weighting.**
(TIF)

**S1 Table. Description of data sources.**
(DOCX)

**S2 Table. Definition of outcomes according to ICD-9 and ICD-10 from the Med-Echo databases.**
(DOCX)

**S3 Table. Risk score definition of $CHA_2DS_2$-VASc and modified HAS-BLED.**
(DOCX)

**S4 Table. Definition of variables used in the risk score definition of $CHA_2DS_2$-VASc according to the ICD-9 and ICD-10 codes from the Med-Echo databases.**
(DOCX)

**S5 Table. Definition of variables used in the risk score definition of HAS-BLED based on the associated morbidities and concomitant drugs used.**
(DOCX)

**S6 Table. Medication with pharmacokinetic drug interactions with anticoagulants after inverse probability of treatment weighting.**
(DOCX)

**S7 Table. Initial cohort and cohort after inverse probability of treatment weighting in low-dose rivaroxaban (15 mg) and standard-dose (20 mg) groups.**
(DOCX)

**S8 Table. Initial cohort and cohort after inverse probability of treatment weighting in low-dose apixaban (2.5 mg) and standard-dose (5 mg) groups.**
(DOCX)

**S9 Table. Effectiveness and safety outcomes in the under treatment cohort after inverse probability treatment weighting.**
(DOCX)

**S10 Table. Effectiveness and safety outcomes in the intent-to-treat cohort after inverse probability of treatment weighting.**
(DOCX)

**S11 Table. Sensitivity analysis of negative controls after inverse probability of treatment weighting.**
(DOCX)

**S12 Table. E-values for significant comparisons between low-dose and standard-dose rivaroxaban and apixaban groups under treatment after inverse probability of treatment weighting.**
(DOCX)

**S13 Table. Distribution of follow-up time and level of adherence at 1-year follow-up.**
(DOCX)

**S14 Table. Sample size estimation.**
(DOCX)

**S15 Table. Baseline characteristics after propensity score matching (1:1).**
(DOCX)

**S16 Table. Comparative effectiveness and safety outcomes in propensity-score-matched cohorts on-treatment (UT) and intent-to-treat (ITT).**
(DOCX)

**S17 Table. Effectiveness and safety outcomes in the under treatment cohort from raw data, IPTW method and PS match 1:1.**
(DOCX)

## Acknowledgments

We would like to thank the RAMQ and Quebec Health Ministry for assistance in handling the data and the Commission d'accès à l'information for authorizing the study.

## Author Contributions

**Conceptualization:** Sylvie Perreault, Robert Côté, Alice Dragomir, Brian White-Guay.

**Formal analysis:** Marc Dorais.

**Funding acquisition:** Sylvie Perreault.

**Methodology:** Sylvie Perreault, Robert Côté, Alice Dragomir.

**Project administration:** Sylvie Perreault.

**Supervision:** Sylvie Perreault, Marc Dorais.

**Validation:** Sylvie Perreault, Robert Côté, Alice Dragomir, Brian White-Guay, Aurélie Lenglet, Marc Dorais.

**Visualization:** Marc Dorais.

**Writing – original draft:** Sylvie Perreault.

**Writing – review & editing:** Sylvie Perreault, Robert Côté, Alice Dragomir, Brian White-Guay, Aurélie Lenglet, Marc Dorais.

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
