## [Decision Letter · Decision Letter 0]

17 Jan 2022

PONE-D-21-29130Effectiveness and safety of low-dose versus high-dose rivaroxaban and apixaban in patients with atrial fibrillationPLOS ONE

Dear Dr. Perreault,

Thank you for submitting your manuscript to PLOS ONE. After careful consideration, we feel that it has merit but does not fully meet PLOS ONE’s publication criteria as it currently stands. Therefore, we invite you to submit a revised version of the manuscript that addresses the points raised during the review process.

 Please submit your revised manuscript by Feb 28 2022 11:59PM. If you will need more time than this to complete your revisions, please reply to this message or contact the journal office at plosone@plos.org. Please include the following items when submitting your revised manuscript:A rebuttal letter that responds to each point raised by the academic editor and reviewer(s). You should upload this letter as a separate file labeled 'Response to Reviewers'.A marked-up copy of your manuscript that highlights changes made to the original version. You should upload this as a separate file labeled 'Revised Manuscript with Track Changes'.An unmarked version of your revised paper without tracked changes. You should upload this as a separate file labeled 'Manuscript'.

We look forward to receiving your revised manuscript.

Kind regards,

Nienke van Rein

Academic Editor

PLOS ONE

Journal Requirements:

No competing interests

Reviewers' comments:

Reviewer's Responses to Questions

**Comments to the Author**

1. Is the manuscript technically sound, and do the data support the conclusions?

Reviewer #1: No

Reviewer #2: Partly

2. Has the statistical analysis been performed appropriately and rigorously? 

Reviewer #1: No

Reviewer #2: Yes

3. Have the authors made all data underlying the findings in their manuscript fully available?

Reviewer #1: No

Reviewer #2: No

4. Is the manuscript presented in an intelligible fashion and written in standard English?

Reviewer #1: Yes

Reviewer #2: No

5. Review Comments to the Author

Reviewer #1: The manuscript has some significant flaws in design and execution which make it challenging to review:

The appropriate dosage of DOAC (Low/high) is determined by clinical characteristics, such as age and renal function

The study design ignores this issue completely- and to my understanding this is a major flaw.

Ignoring the dosage labeling may create significant biases.

It is impossible to estimate how the lower effectiveness is a result of a lower dose, or perhaps an inappropriate dose.

Even if we ignore this flaw, the use of IPTW is not sufficient and not sure it is appropriate for the study objective.

For example, the age of the low dose apixaban before IPTW is 10 years higher than the high dose- suggesting that the low dose maybe a result of the age of the patients.

I would expect to see the real (raw) number of events in each group.

The data is disclosed in Table S8, is after IPTW, and does not provide this info.

At the least, I would expect using an alternative method for assessing effectiveness to validate the IPTW modeling.

The abundance of data should be enable these analyses

Reviewer #2: Perreault et al used Canadian administrative databases to study the effectiveness and safety of low dose vs high dose rivaroxaban and apixaban in patients with atrial fibrillation. The authors found no clear differences in effect or safety comparing high with low dose of either drug.

I have a number of questions/comments regarding the study’s methods that should be answered before evaluation. Essentially, I would like to see a clearer description of the Methods section.

1. The authors used data from the “Med-Echo” databases. For authors should include a description of these data sources as most readers will be unaware of the infrastructure of these data sources. A few additional lines could be added to the main text, with the rest included in the Appendix. As of now, no description is present, even in the Appendix.

2. The authors should make the cohort entry date (or index date) more explicit for the reader. I suspect this data to be the date of drug filling, but it is not clear. Also, please be consistent regarding terminology (cohort entry vs index date). Relatedly, it would be nice if the authors included a study diagram, which makes it easier for readers to understand the study (see https://doi.org/10.7326/M18-3079)

3. The authors used a quite long exposure assessment window. Can the authors provide the median time to initiation by exposure group? What happened if an AF patient died (or experienced a stroke) before DOAC initiation?

4. Please describe the exposures (ie. low vs high) more clearly. Which specific drugs were classified as low and which as high?

5. Please provide a rationale for the exclusion criteria.

6. Why was IPTW used over other weighting methods? What question was trying to be answered?

7. The authors include a range of covariates in their model, but the covariates are not clearly described before Table

6. PLOS authors have the option to publish the peer review history of their article (what does this mean?). If published, this will include your full peer review and any attached files.

Reviewer #1: No

Reviewer #2: No

---

## [Author Response · Author response to Decision Letter 0]

8 Mar 2022

Data Availability statement: All relevant data are within the manuscript and its Supporting Information files.

Captions for our Supporting Information files were added at the end of the manuscript.

Role of Funder statement was added in our amended cover letter (dated March 8, 2022).

The sources of funding were clarified in our amended cover letter (dated March 8, 2022).

The title of the manuscript was amended on the online submission form so that it is identical to the one indicated in the Manuscript. 

We also amended from the ''Running title'' the ''versus high-dose'' for ''versus standard-dose'' as indicated in the full title of the manuscript. This was amended on the online submission form and also in the Manuscript.

The authors have declared that no competing interests exist.

---

## [Decision Letter · Decision Letter 1]

10 Oct 2022

PONE-D-21-29130R1Effectiveness and safety of low-dose versus standard-dose rivaroxaban and apixaban in patients with atrial fibrillationPLOS ONE

Dear Dr. Perreault,

Thank you for submitting your manuscript to PLOS ONE. After careful consideration, we feel that it has merit but does not fully meet PLOS ONE’s publication criteria as it currently stands. Therefore, we invite you to submit a revised version of the manuscript that addresses the points raised during the review process.

ACADEMIC EDITOR: 

Dear Authors,

Please answer to the issues raised by the reviewer. In case the issue cannot be solved by a reanalysis of the data, the limitation should be clearly stated in the revised manuscript.

We look forward to receiving your revised manuscript.

Kind regards,

Roberto Magalhães Saraiva, MD, PhD

Academic Editor

PLOS ONE

Additional Editor Comments (if provided):

Dear Authors,

Please answer to the issues raised by the reviewer. In case the issue cannot be solved by a reanalysis of the data, the limitation should be clearly stated in the revised manuscript.

Reviewers' comments:

Reviewer's Responses to Questions

**Comments to the Author**

1. If the authors have adequately addressed your comments raised in a previous round of review and you feel that this manuscript is now acceptable for publication, you may indicate that here to bypass the “Comments to the Author” section, enter your conflict of interest statement in the “Confidential to Editor” section, and submit your "Accept" recommendation.

Reviewer #1: (No Response)

Reviewer #2: All comments have been addressed

2. Is the manuscript technically sound, and do the data support the conclusions?

Reviewer #1: No

Reviewer #2: Yes

3. Has the statistical analysis been performed appropriately and rigorously? 

Reviewer #1: No

Reviewer #2: Yes

4. Have the authors made all data underlying the findings in their manuscript fully available?

Reviewer #1: Yes

Reviewer #2: Yes

5. Is the manuscript presented in an intelligible fashion and written in standard English?

Reviewer #1: Yes

Reviewer #2: Yes

6. Review Comments to the Author

Reviewer #1: In short, the response to the effectiveness evaluation methods (my third comment) is satisfactory.

However, the answers to my first 2 comments (Although very details) do not answer my very basic and important comment regarding the appropriateness of dosage, according to FDA dosage labels for each drug.

I do not think that this issue can be answered only by the literature, but has to be analyzed by the data.

Even if the authors disagree to make it the primary analysis, it should be done as a secondary or sensitivity analysis.

Reviewer #2: The authors have adequately addressed my comments related to the methods section. I have no further concerns.

7. PLOS authors have the option to publish the peer review history of their article (what does this mean?). If published, this will include your full peer review and any attached files.

Reviewer #1: No

Reviewer #2: No

---

## [Author Response · Author response to Decision Letter 1]

26 Oct 2022

PONE-D-21-29130R1

Effectiveness and safety of low-dose versus standard-dose rivaroxaban and apixaban in patients with atrial fibrillation

Please answer to the issues raised by the reviewer. In case the issue cannot be solved by a reanalysis of the data, the limitation should be clearly stated in the revised manuscript.

Many thanks for the excellent comments which improve the clarity of the manuscript. 

Comments to the Reviewer 1

Reviewer #1: In short, the response to the effectiveness evaluation methods (my third comment) is satisfactory.

However, the answers to my first 2 comments (Although very details) do not answer my very basic and important comment regarding the appropriateness of dosage, according to FDA dosage labels for each drug. I do not think that this issue can be answered only by the literature, but has to be analyzed by the data. Even if the authors disagree to make it the primary analysis, it should be done as a secondary or sensitivity analysis.

In response: As reported in the limits section, this is an observational study using administrative data and may be subject to confounding bias by unadjusted factors (e.g., AF severity, blood pressure control, exact estimated glomerular filtration rate, international normalized ratio, body weight, and over-the-counter prescriptions) or by a residual channeling bias. 

We are not able to provide the appropriateness of the prescription without having the exact weight and the exact estimated glomerular filtration rate. But, for chronic kidney disease, we have a validated algorithm for the stage of chronic kidney disease. The category of estimated glomerular filtration rate was estimated with an algorithm based on diagnosis code, drug use, and nephrologist visits from administrative databases that shown to be valid when compared with medical chart reviews in older adults (Roy L et al. Can J Kidney Health Dis 2020). The algorithm used for eGFR definition had a positive predictive value ranging from 94.5% to 97.7% (Roy L et al 2020). We modified the method section at Page 7, line 140-144 as followed: The category of chronic kidney disease was assessed using the estimated glomerular filtration rate using an algorithm based on diagnosis code, drug use, and nephrologist visits from administrative databases that shown to be valid when compared with medical chart reviews in older adults. The algorithm used for eGFR definition had a positive predictive value ranging from 94.5% to 97.7%

In order to assess the robustness of the results we provided several sensitivity analyses to support our hypothesis that given that the use of low-dose DOAC is more prevalent in clinical practice, and that close to 50% of patients who receive the low-dose apixaban do not meet the least two of three clinical characteristics (de Vries TAC et al. 2020), it is important to assess the effectiveness and safety of low-dose vs standard-dose DOACs used in AF patients with similar baseline characteristics (Maura G et al, 2017; Perreault S et al 2022). Due to the lack of randomization in treatment assignment, we used an inverse probability treatment weighting (IPTW) approach to minimize the impact of confounding by indication in representative cohorts of AF patients to assess the effectiveness and safety of low-dose vs standard-dose rivaroxaban and apixaban (Allan V et al, 2020).

We modified the limit section at Page 26, Para 307-309, as followed: Second, we are not able to provide the appropriateness of the prescription without having the exact weight and the exact estimated glomerular filtration rate but in order to assess the robustness of the results, we ran several sensitivity analyses.

Reviewer #2: The authors have adequately addressed my comments related to the methods section. I have no further concerns.

---

## [Decision Letter · Decision Letter 2]

3 Nov 2022

Effectiveness and safety of low-dose versus standard-dose rivaroxaban and apixaban in patients with atrial fibrillation

PONE-D-21-29130R2

Dear Dr. Perreault,

We’re pleased to inform you that your manuscript has been judged scientifically suitable for publication and will be formally accepted for publication once it meets all outstanding technical requirements.

Kind regards,

Roberto Magalhães Saraiva, MD, PhD

Academic Editor

PLOS ONE

Additional Editor Comments (optional):

Reviewers' comments:

Reviewer's Responses to Questions

**Comments to the Author**

1. If the authors have adequately addressed your comments raised in a previous round of review and you feel that this manuscript is now acceptable for publication, you may indicate that here to bypass the “Comments to the Author” section, enter your conflict of interest statement in the “Confidential to Editor” section, and submit your "Accept" recommendation.

Reviewer #1: All comments have been addressed

Reviewer #2: All comments have been addressed

2. Is the manuscript technically sound, and do the data support the conclusions?

Reviewer #1: (No Response)

Reviewer #2: Yes

3. Has the statistical analysis been performed appropriately and rigorously? 

Reviewer #1: (No Response)

Reviewer #2: Yes

4. Have the authors made all data underlying the findings in their manuscript fully available?

Reviewer #1: (No Response)

Reviewer #2: Yes

5. Is the manuscript presented in an intelligible fashion and written in standard English?

Reviewer #1: (No Response)

Reviewer #2: Yes

6. Review Comments to the Author

Reviewer #1: (No Response)

Reviewer #2: (No Response)

7. PLOS authors have the option to publish the peer review history of their article (what does this mean?). If published, this will include your full peer review and any attached files.

Reviewer #1: No

Reviewer #2: No

---

## [Editor Report · Acceptance letter]

21 Nov 2022

PONE-D-21-29130R2 

Effectiveness and safety of low-dose versus standard-dose rivaroxaban and apixaban in patients with atrial fibrillation 

Dear Dr. Perreault:

I'm pleased to inform you that your manuscript has been deemed suitable for publication in PLOS ONE. Congratulations! Your manuscript is now with our production department. 

Kind regards, 

on behalf of

Dr. Roberto Magalhães Saraiva 

Academic Editor

PLOS ONE